# MAD-TD: MODEL-AUGMENTED DATA STABILIZES HIGH UPDATE RATIO RL

**Claas A Voelcker**[*]
University of Toronto
Vector Institute

**Marcel Hussing**[*]
University of Pennsylvania

**Eric Eaton**
University of Pennsylvania

**Amir-massoud Farahmand**
Polytechnique Montréal
Mila – Quebec AI Institute
University of Toronto

**Igor Gilitschenski**
University of Toronto
Vector Institute

## ABSTRACT

Building deep reinforcement learning (RL) agents that find a good policy with few samples has proven notoriously challenging. To achieve sample efficiency, recent work has explored updating neural networks with large numbers of gradient steps for every new sample. While such high update-to-data (UTD) ratios have shown strong empirical performance, they also introduce instability to the training process. Previous approaches need to rely on periodic neural network parameter resets to address this instability, but restarting the training process is infeasible in many real-world applications and requires tuning the resetting interval. In this paper, we focus on one of the core difficulties of stable training with limited samples: the inability of learned value functions to generalize to unobserved on-policy actions. We mitigate this issue directly by augmenting the off-policy RL training process with a small amount of data generated from a learned world model. Our method, Model-Augmented Data for Temporal Difference learning (MAD-TD) uses small amounts of generated data to stabilize high UTD training and achieve competitive performance on the most challenging tasks in the DeepMind control suite. Our experiments further highlight the importance of employing a good model to generate data, MAD-TD's ability to combat value overestimation, and its practical stability gains for continued learning.

## 1 INTRODUCTION

Instead of solely relying on data gathered by a target policy, *off-policy* reinforcement learning (RL) aims to leverage experience gathered by past policies (Sutton & Barto, 2018) to fit a value function for the target policy. Ideally, training over many iterations should help extract important information from past data. However, recent work has shown that simply increasing the number of gradient update steps, the *replay ratio* or *update-to-data (UTD) ratio*, can lead to highly unstable learning (Nikishin et al., 2022; D'Oro et al., 2023; Hussing et al., 2024; Nauman et al., 2024b).

Prior work has stabilized the learning by using double Q minimization to reduce overestimation (Fujimoto et al., 2018), training ensemble methods to improve value estimation (Chen et al., 2020; Hiraoka et al., 2022), introducing architectural regularization (Hussing et al., 2024; Nauman et al., 2024b), or resetting networks periodically throughout the learning process (D'Oro et al., 2023; Schwarzer et al., 2023; Nauman et al., 2024b). However, while useful, pessimistic underestimation and architectural regularization are insufficient by themselves to combat the problem (Hussing et al., 2024), and so most methods resort to either network resets or ensembles. Critic ensembles can be expensive to train, and resetting has several important limitations: in real systems, re-executing a random policy can be expensive or unsafe, the resetting interval needs to be carefully tuned (Hussing et al., 2024), and when storing a full reset buffer is infeasible, resetting loses important information.

We narrow in on a key issue contributing to unstable training: *wrong value function estimation on unobserved on-policy actions* (Thrun & Schwartz, 1993; Tsitsiklis & Van Roy, 1996). Off-policy

RL uses the values of states sampled under old policies with actions from the target policy to update the value function. However, these state-action pairs themselves are not in the replay buffer and hence their value estimate is not directly improved by training. Consequently, a learned function which achieves low error on seen data is not guaranteed to generalize well to actions that *differ* from past actions. This problem is related to overfitting (Li et al., 2023) and contributes to overestimation (Thrun & Schwartz, 1993; Hasselt, 2010; Fujimoto et al., 2018). However, overfitting assumes that train and test set are drawn from the same distribution, while we focus on the distribution shift between data collection and target policy. Previous work has investigated the difficulty of off-policy learning due to this shift (Maei et al., 2009; Sutton et al., 2016; Hasselt, 2010; Fujimoto et al., 2018), yet there are no tractable mitigation strategies that work well in the high UTD regime with deep RL.

To corroborate our hypothesis that generalization to unobserved actions is a major obstacle for training at high UTDs, we examine the behavior of value functions on on-policy transitions. Our experiments reveal that value functions generalize significantly worse to unobserved on-policy action transitions than to validation data from the same distribution as the training set. Building on this, we propose to improve on-policy value estimation by using *model-generated on-policy data*.

Previous investigations into model-based deep RL have focused on learning values fully in model roll-outs (Buckman et al., 2018; Janner et al., 2019; Hafner et al., 2020; Ghugare et al., 2023) and the associated challenges (Zhao et al., 2023; Hansen et al., 2024). In contrast, we show that mixing a small amount of model-generated on-policy data with real off-policy replay data is sufficient to achieve stable learning in the high UTD regime. Our method, Model-Augmented Data for Temporal Difference learning(MAD-TD), mitigates the generalization issues of the value function in the hardest tasks of the DeepMind control (DMC) benchmark (Tunyasuvunakool et al., 2020b) and achieves strong and stable high UTD learning without resetting or redundant ensemble learning.

The main contributions of this work are twofold: First, we empirically show the existence of misgeneralization from off-policy value estimation to on-policy predictions. We connect this issue to the challenge of stable learning with high update ratios and highlight how increasing the update ratio increases Q function overestimation. Second, we provide a new method called MAD-TD that improves the value function accuracy on unobserved on-policy actions with model-generated data and stabilizes training at high update ratios. This method proves to have equivalent performance to or outperform previous strong baselines.

## 2 MATHEMATICAL BACKGROUND

We consider a standard RL setting, the discounted infinite-horizon MDP $(\mathcal{X}, \mathcal{A}, \mathcal{P}, r, \rho, \gamma)$ with state space $\mathcal{X}$, action space $\mathcal{A}$, a transition kernel $\mathcal{P} : \mathcal{X} \times \mathcal{A} \to \mathcal{M}(\mathcal{X})$, a reward function $r : \mathcal{X} \times \mathcal{A} \to \mathbb{R}$, starting state distribution $\rho \in \mathcal{M}(\mathcal{X})$ and a discount factor $\gamma \in [0, 1)$ (Puterman, 1994; Sutton & Barto, 2018). For a space $Y$ we use $\mathcal{M}(Y)$ to denote the set of probability measures over the space. Our goal is to learn a policy $\pi : \mathcal{X} \to \mathcal{M}(\mathcal{A})$ that maximizes the discounted sum of future rewards

$$\pi^* \in \arg\max_{\pi \in \Pi} \sum_{t=0}^{\infty} \mathbb{E}_{\mathcal{P}^\pi}[\gamma^t r(x_t, a_t)|x_0 \sim \rho] \;, \tag{1}$$

where actions are sampled according to the policy and new states according to the transition kernel.

### 2.1 OFF-POLICY VALUE FUNCTION LEARNING

As an intermediate objective, many algorithms attempt to simplify the direct policy optimization problem by first learning a policy value function $Q^\pi$, which is defined via a recursive equation

$$Q^\pi(x, a) = r(x, a) + \gamma \mathbb{E}_{x' \sim \mathcal{P}(\cdot|x,a), a' \sim \pi(\cdot|x')}[Q^\pi(x', a')] \;. \tag{2}$$

The policy can then be incrementally improved by picking $\pi_{k+1}(x) \in \arg\max_{a \in \mathcal{A}} Q^{\pi_k}(x, a)$ at every time step $k$. In practice, $Q^\pi$ and $\pi$ are often parameterized as neural networks and learned from data. To increase the sample efficiency of the algorithm, it is common to store *all* collected interaction data independent of the collection policy in a replay buffer $\mathcal{D} = \{(x_t, a_t, r_t, x_{t+1})_{t=0}^T\}$. As the Q-value only depends on the policy via the policy evaluation at the next state, it is possible

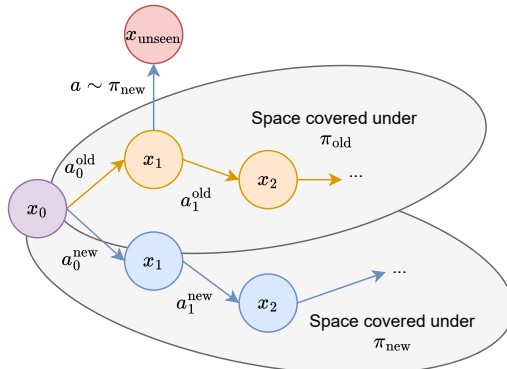

Figure 1: A visualization of the core issue we investigate. Even if a replay buffer contains good coverage for two policies ($\pi_{\text{old}}$ and $\pi_{\text{new}}$) starting from $\rho = x_0$, this does not guarantee that it contains a transition for executing an action under the new policy on a state visited under the old. However, this state-action pair's value estimate is used to update the value of state $x_0$ via Equation 3, without being grounded in an observed transition.

to estimate Q-values from past interaction data by minimizing the fitted Q-learning objective

$$\mathcal{L}\left(\hat{Q}\Big|\mathcal{D},\pi\right) = \frac{1}{|\mathcal{D}|}\sum_{t=0}^{T}\left|\hat{Q}(x_t,a_t) - \left[r_t + \gamma\hat{Q}\left(x_{t+1},a'\right)\right]_{\text{sg}}\right|^2 \quad \text{with } a' \sim \pi(\cdot|x_{t+1}) . \quad (3)$$

Here $[\cdot]_{\text{sg}}$ denotes the stop gradient operation introduced to avoid the double sampling bias and all data contained in the replay buffer is colored blue. However, the Q value at the next state $x_{t+1}$ is evaluated with an action $a'$ that is *not* guaranteed to be in the replay memory, as the target policy can be different from the policy used to gather the sample. This means that we require the Q value to generalize to potentially unseen actions. We provide a visualization of this issue in Figure 1.

## 3 INVESTIGATING THE ROOT CAUSE OF UNSTABLE Q LEARNING

Minimizing Equation 3 finds the policy Q function over a replay buffer with sufficient coverage of all states and actions that this policy visits. However, in most continuous control RL algorithms (Lillicrap et al., 2016; Haarnoja et al., 2018; Fujimoto et al., 2018), this update is interleaved with policy update steps . The data in $\mathcal{D}$ then necessarily becomes *off-policy* as training progresses.

This means that the number of actor and critic optimization steps needs to be balanced with gathering new data. Obtaining new on-policy data is vital to continually improve policy performance (Ostrovski et al., 2021), but performing more update steps before gathering new data ensures that the existing data has been used effectively to improve the policy. The *replay ratio* (Fedus et al., 2020) or *update-to-data (UTD) ratio* (Nikishin et al., 2022), which governs the number of gradient steps per environment step, is therefore a vital hyperparameter.

Naively training with high UTD ratios can lead to collapse in off-policy deep RL (Nikishin et al., 2022). We conjecture that one of the major causes of the instability of high UTD off-policy learning are wrong Q values on *unobserved actions*. This is a well-known problem for off-policy TD learning (Baird, 1995; Tsitsiklis & Van Roy, 1996; Sutton et al., 2016; Ghosh & Bellemare, 2020). To differentiate the problem from *overfitting* to the training distribution, we use the term *misgeneralization* to highlight the importance of the distribution shift in causing the issue. Our experiments in Subsection 3.2 show that generalization to on-policy actions is more difficult than generalization to a validation dataset that follows the training distribution, and that higher UTDs exacerbate the issue.

### 3.1 ACTION DISTRIBUTION SHIFT CAN CAUSE OFF-POLICY Q VALUE DIVERGENCE

To highlight the role that on-policy actions play in stabilizing Q value learning, we show an analysis the stability of Q learning with linear features. The core ideas follow Sutton et al. (2016) and are also explored by Tsitsiklis & Van Roy (1996); Sutton (1988). We assume that the Q function is parameterized with fixed features and weights as $Q(x,a) = \phi(x,a)^\top\theta$. Let $X$ and $A$ be the sizes of the state and action space respectively. Let $P \in \mathbb{R}^{X\cdot A\times X}$ be the matrix of transition probabilities from state-action pairs to states. A policy can then be expressed as a mapping $\Pi \in \mathbb{R}^{X\times X\cdot A}$ from states to the likelihood of taking each action. $R \in \mathbb{R}^{X\cdot A}$ is the vector of rewards. $D^\pi \in \mathbb{R}^{X\cdot A\times X\cdot A}$ is a matrix where the main diagonal contains the discounted state-action occupancies of $P^\pi$ starting

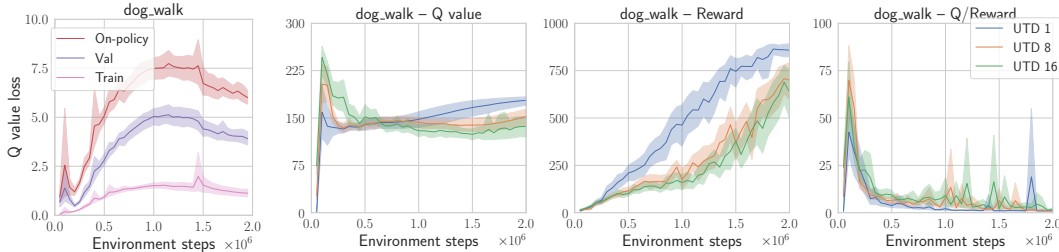

Figure 2: Left: the train, validation, and on-policy validation error of the Q function at UTD 1. Right: the Q values and return curves of TD3 agents across different UTD 1, 8, and 16.

from $\rho$. If we assume access to a mixed replay buffer $\mathcal{D} = \bigcup \{D^{\pi_1}, \ldots, D^{\pi_n}\}$ gathered with different policies, the Q learning loss for a target policy $\Pi$ can be written as

$$L(\theta) = \sum_{i=1}^{n} \left[ D^{\pi_i} \left( \Phi^\top \theta - [R + \gamma P \Pi \Phi^\top \theta]_{\text{sg}} \right)^2 \right] \quad . \tag{4}$$

The stability of learning with this loss can be analyzed using the gradient flow

$$\dot{\theta} = -2\Phi \sum_{i=1}^{n} D^{\pi_i} \left( I - \gamma P \Pi \right) \Phi^\top \theta + 2\Phi \sum_{i=1}^{n} D^{\pi_i} R \quad . \tag{5}$$

This gradient flow is guaranteed to to be stable around a fixed point $\theta^*$ if the key matrix $\sum_{i=1}^{n} D^{\pi_i} (I - \gamma P \Pi)$ is positive definite (Sutton, 1988). Details and a proof of the following statement are provided in Appendix C. We can decompose the key matrix and see that the positive definiteness depends on the difference in policy between the replay buffer and the target policy

$$\sum_{i=1}^{n} D^{\pi_i} \left( I - \gamma P \Pi \right) = \underbrace{\sum_{i=1}^{n} D^{\pi_i} \left( I - \gamma P \Pi_i \right)}_{\text{positive definite}} + \gamma \underbrace{\sum_{i=1}^{n} D^{\pi_i} P (\Pi_i - \Pi)}_{\text{no guarantees}} \quad . \tag{6}$$

In general, we can provide no guarantees for the second term outside of the on-policy case ($\Pi_i = \Pi$) where it becomes $0$. The stability depends on the difference between the target policy and the data-collection policies. If the target policy takes actions which are not well covered under the training policies, the remainder can be non positive definite. This also matches the intuition that learning fails if we simply do not have sufficient evidence for the Q function of unobserved actions.

When using features, the eigenvalue conditions on the key matrix are only sufficient, not necessary, as the features can allow for sufficient generalization between observed and unobserved state-action pairs. In deep RL, the features $\phi$ are updated alongside with the weights, making it hard to provide definitive mathematical statements on stability. With good function approximation, we could hope that the learned value function generalizes correctly to unseen actions. In the next section we investigate this for a non-trivial task from the DMC suite and highlight that, while the value function does not diverge irrecoverably, good generalization is not guaranteed either.

### 3.2 EMPIRICAL Q VALUE ESTIMATION WITH OFF-POLICY DATA

In environments with large state-action space, ensuring coverage is difficult. To investigate whether learning is stable nonetheless, we train a model-free TD3 agent on the *dog walk* environment (Tunyasuvunakool et al., 2020a). The architecture is presented in Subsection 4.1, and is regularized to prevent catastrophic divergence (Hussing et al., 2024; Nauman et al., 2024a) and uses clipped double Q learning (Fujimoto et al., 2018). This means it uses the most common techniques which are designed to prevent misgeneralization and overestimation.

While training a TD3 agent (Fujimoto et al., 2018), we save transitions in a validation buffer with a 5% probability. At regular intervals we compute the critic loss on this validation set. In addition, we reset our simulator to each validation state and sample an action from the target policy. We then

simulate the ground truth on-policy transition and compute the loss over these. This allows us to test how well our value function generalizes to target policy state-action pairs (as depicted in Figure 1).

The results are presented in Figure 2 and show a gap both between the train and validation sets, as well as the validation and the on-policy sets. While we use the on-policy state-actions to update the Q value, these estimates are not actually consistent with the environment. Furthermore, the Q value overestimation grows with increasing UTDs. This phenomenon was previously discussed in the context of over-training on limited data (Hussing et al., 2024) .

The experiments show that the problem outlined in Subsection 3.1 is not merely a mathematical curiosity, but that Q value generalization to out-of-replay-distribution actions is difficult in practice, and becomes more difficult with increasing update ratios. Even though full divergence is not observed as new data is continually added to the replay buffer, it takes a long time for the effects of severe early overestimation to dissipate.

## 3.3 PREVIOUS ATTEMPTS TO COMBAT MISGENERALIZATION AND OVERESTIMATION

Prior strategies that deal with misgeneralization can be grouped into three major directions: architectural regularization to prevent divergence of the value function, pessimism or ensemble learning to combat overestimation, and networks resets to restart learning. While all of these interventions help to some degree, they each either do not solve the problem in full or cause additional issues. We outline highly related work here and provide an additional related work section in Appendix B.

**Architectural regularization** Architecture changes (Hussing et al., 2024; Nauman et al., 2024a;b; Lyle et al., 2024) and auxiliary feature learning losses (Schwarzer et al., 2021; Zhao et al., 2023; Ni et al., 2024; Voelcker et al., 2024) are largely reliable interventions, and have shown to provide improvements without much drawbacks in prior work. However, as Hussing et al. (2024) and our experiment presented in Subsection 3.2 highlight, by themselves they can mitigate catastrophic overestimation and divergence, but do not guarantee proper generalization.

**Pessimism and ensembles** To combat overestimation directly, the most prominent approach in continuous action spaces is Clipped Double Q Learning (Fujimoto et al., 2018). Here, a Q value estimate is obtained from two independent estimates $\hat{Q}_1$ and $\hat{Q}_2$. If the error of the two critic estimators is assumed to be independent noise on the true critic estimate then using the minimum over both estimates is guaranteed to underestimate the true critic value in expectation. However, in complex settings this assumption on the the error of the critic estimates may not hold.

Ensembles (Lan et al., 2020; Chen et al., 2020; Hiraoka et al., 2022; Farebrother et al., 2023) or online tuning of the rate of pessimism (Moskovitz et al., 2021) have been proposed to obtain tighter lower bounds on the Q value. However, these strategies can be expensive as redundant models or hyperparameter tuning are needed. As a simpler strategy, recent works have also employed clipping to obtain an upper bound of the Q function to prevent divergence (Fujimoto et al., 2024).

**Resetting** Finally, network resets been shown to mitigate training problems (Nikishin et al., 2022; D'Oro et al., 2023; Schwarzer et al., 2023; Nauman et al., 2024b) in high UTD regimes. However, in cases where the agent fails to explore any useful parts of the state space within the reset interval, restarting the learning process will not improve performance (Hussing et al., 2024). This makes tuning the resetting interval both important and potentially difficult and no tuning recipes have been presented. Resetting is also a potentially hazardous strategy in real-world applications, where re-executing a random policy might be costly or infeasible due to safety constraints. Finally, it heavily relies on the assumption that all past interaction data can be kept in the replay buffer.

**Data generation** Lu et al. (2024) attempts to combat failures of high UTD learning by supplementing a replay buffer with data generated from a trained diffusion model. This idea is inspired by the hypothesis that failure to learn in high-UTD settings is caused by a lack of data (Nikishin et al., 2022). The method, SynthER, improves learning accuracy on simple tasks in the DMC benchmark. However, we demonstrate that simply adding more data is insufficient to combat misgeneralization by comparing SynthER to MAD-TD in Appendix B and Subsection E.4.

All of these strategies are somewhat able to alleviate the problem of out-of-distribution value estimation, yet none of them directly address the issue at the root. In the next chapter, we present an alternate approach that aims to directly regularize the action value estimates under the target policy.

## 4 MITIGATION VIA MODEL-GENERATED SYNTHETIC DATA

As value functions misgeneralize due to lack of sufficient on-policy data, we propose to obtain synthetic data from a learned model instead. However, model-based RL can also cause problems such as compounding world model errors and optimistic exploitation of errors in the learned model. By using both real and model-generated data, we can trade-off these issues: on-policy data improves the value function and limits the impact of off-policy distribution shifts, while using only a limited number of model-generated samples prevents model errors from deteriorating the value estimates.

Our approach builds on the TD3 algorithm (Fujimoto et al., 2018) and uses an update ratio of 8 by default. Our critic is updated with both model-based and real data following the DYNA framework (Sutton, 1990). More precisely, we replace a small fraction $\alpha$ of samples $\{x, a, r, x'\}$ in each batch with samples from a learned model $\hat{p}$ starting from the same state $\{x, \pi(x), \hat{r}, \hat{x}'\}$ with $\hat{r}, \hat{x}' \sim \hat{p}(\cdot|x, \pi(x))$. In our experiments, $\alpha$ is set to merely 5%. We found that this small amount provides competitive performance across a wide range of values (compare Subsection E.3). We term this approach Model-Augmented Data for Temporal Difference learning (MAD-TD).

**Model vs Q function generalization** We expect that a learned models will yield better generalization than the Q function for two reasons. First, the policy is updated each step to find an action that maximizes the value function. This means we are effectively conducting an adversarial search for overestimated values. The model's reward and state estimation error on the other hand are independent of this process. We test the adversarial robustness of our model-augmented value functions in Subsection 5.3. Second, our experiment shows that value functions primarily diverge at the beginning of training. In these cases, coverage is low and on-policy state-action pairs are often not available. Obtaining a slightly wrong, yet converging value estimate can then be more useful than a diverging one. Even as more data is gathered, new policies might not revisit old states with a high likelihood. Therefore even as training continues we expect the model data to provide some benefit.

### 4.1 DESIGN CHOICES AND TRAINING SETUP

Our model is based on the successful TD-MPC2 model (Hansen et al., 2024) combined with the deterministic actor-critic algorithm TD3 (Fujimoto et al., 2018). We aim to reduce the complexity of TD-MPC2 to the minimal necessary components to achieve strong learning in the DM Control suite, and thus forgo added exploration noise, SAC, ensembled critics, and longer model rollout for training or policy search. We outline several design choices here and refer to Section D for more detail. We additionally ablate our version of the model against TD-MPC2 in Subsection E.5.

**Encoder:** Like TD-MPC2, we parameterize the state with a learned encoder $\phi : \mathcal{X} \to \mathcal{Z}$ with a SimNorm nonlinearity (Lavoie et al., 2023). This transformation groups a latent vector into groups of $k$ entries and applies a softmax transformation over each group. This bounds the norm of the features, which has been shown aid with stable training (Hussing et al., 2024; Nauman et al., 2024a).

**Critic representation and loss:** We use the HL-Gauss transformation to represent the Q function (Farebrother et al., 2024). The critic loss is the cross-entropy between the estimated Q function's categorical representation and the bootstrapped TD estimate. To stabilize learning, we initialize the critic network towards predicting 0 for all states.

**Model loss:** The world model predicts the next state latent representation and the observed reward from a given encoded state $\phi(x)$ and action $a$. The loss has three terms: the cross-entropy loss over the SimNorm representation of the encoded next state, the MSE between the reward predictions, and the cross-entropy between the next state critic estimate and the predicted state's critic estimate. This final term replaces the MuZero loss in TD-MPC2 with a simplified variant based on the IterVAML loss (Farahmand, 2018). We provide the exact mathematical equations for the loss in Appendix D.

**Training:** We train the architecture by interleaving one environment step with one round of updates with a varying number of gradient steps governed by the UTD parameter. For each update step, a new mini-batch is sampled independently from a replay buffer of previously collected experience. We found that varying the number of update steps only for the critic and actor while keeping the update ratio for the model and encoder updates at 1 leads to significantly more stable learning.

**Run-time policy improvement with MPC:** Following the approach outlined by Hansen et al. (2022), the learned model can also be used at planning time to obtain a better policy. Using the model

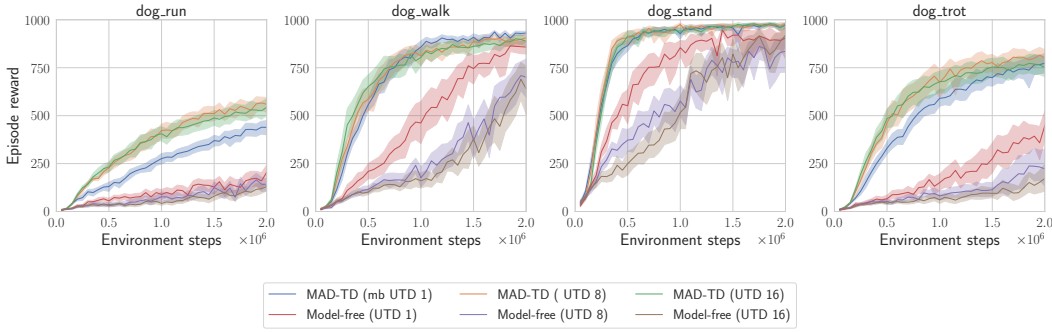

Figure 3: Return curves for the dog tasks with differing UTD values. The return increases or remains stable when training with MAD-TD. Without model data, the performance decreases under high UTD. MPC is turned off in these runs to cleanly evaluate the impact of model data on critic learning.

for MPC at planning time exploits the same benefit of models as the critic learning improvement: we obtain a model-corrected estimate of the value function and choose our policy accordingly. As we only train our model for one step, we also conduct the MPC rollout for one step into the future.

## 5 EXPERIMENTAL EVALUATION

We conduct all of our experiments on the DeepMind Control suite (Tunyasuvunakool et al., 2020b). Following Nauman et al. (2024b)'s recommendations we focus our main comparisons and ablations on the two hardest settings, the *humanoid* and *dog* environments (which we will refer to as the *hard suite*). In Subsection E.8 we furthermore show results for the metaworld benchmark (Yu et al., 2019). Implementation details can be found in Appendix D. Unless stated otherwise we evaluate MAD-TD with a UTD of 8 and use the same hyperparameters across all tasks.

Note that even though we refer to training MAD-TD without using model data for the critic as "model-free", the algorithm still benefits from the model through feature learning which has proven to be a strong regularization technique in high UTD settings (Schwarzer et al., 2023). All main result curves are aggregated across 10 seeds per task. We plot mean and bootstrapped confidence intervals for the mean at the 95% certainty interval. For aggregated plots, we use the library provided by Agarwal et al. (2021). Additional comparisons on more environments are presented in Appendix E.

### 5.1 IMPACT OF USING MODEL-GENERATED DATA

We first repeat the experiment presented in Subsection 3.2 and show the results in Figure 4. Using model-based data closes the gap between on-policy and validation loss. We also observe that the initial Q overestimation disappears, which is consistent across all hard environments (see Subsection E.1). This provides evidence that we are indeed able to overcome the unseen action challenge.

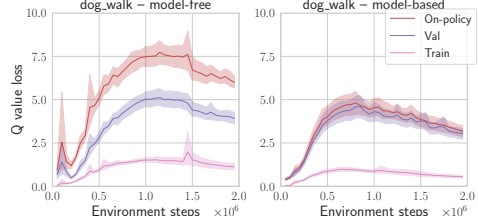

Figure 4: Mean loss values with and without generated data (see Figure 2) for UTD 1.

**Performance with and without model data at varying UTD ratios:** In Figure 3 we present the impact of using model-based data across different UTD ratios. Humanoid results are found in Subsection E.2. As is directly evident, across the dog tasks, we observe stagnating or deteriorating performance when increasing the update ratio, consistent with reports in prior work. However, when using a small fixed amount of model generated data, this trend is reversed across all tested environments, with performance improving or at least remaining consistent. We find that with model-based data, training is stable across a range of UTDs, even beyond those tested in recent high UTD work (Nauman et al., 2024b). We also note that we observe only limited benefits from increasing the UTD ratio when properly mitigating *misgeneralization*, except for the highly challenging dog run task.

**Comparison with baselines:** As our method combines model-free and model-based updates, we compare our method against both TD-MPC2 (Hansen et al., 2024), a strong model-based baseline, and BroNet (Nauman et al., 2024b), a recent algorithm proposed for high UTD learning. Since Nauman et al. (2024b) and Hansen et al. (2024) trained with differing numbers of action repeats, and we found that the performance does not cleanly translate between these regimes, we present our method both with an action repeat value of 1 and 2. Some hyperparameters are adapted to the AR=1 setting (compare Table 2). The results are presented in aggregate in Figure 5, with per environment curves show in Subsection E.6 for hard tasks and Subsection E.7 for a wider range of DMC tasks.. We find that our method performs on par or above previous methods, and strikingly it is able to achieve higher returns faster than both TD-MPC2 and BRO.

## 5.2 PERFORMANCE AND STABILITY IMPACT OF RESETTING

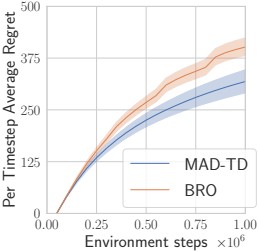

**Resetting comparison:** To investigate if our technique enables more stable training, we set up an experiment to test the effects of resetting on our method. Figure 6 presents aggregate results comparing our approach and BRO, both with and without resetting. Across all tasks we find that resetting barely improves MAD-TDs performance with the tested hyperparameters. Benefits can only be observed on some seeds and can most likely be attributed to restarting the exploration process (Hussing et al., 2024). However, the BRO algorithm is not able to achieve reliable performance without resets. These results highlight that mitigating the problems related to incorrect generalization of the value function stabilize training, and that these problems are likely a major cause of the failure of high UTD learning in the DMC tasks. Conjectured problems like the primacy bias effect (Nikishin et al., 2022) need to be carefully investigated as we do not find evidence that a primacy bias impacts MAD-TD's performance in the DMC environments. Our work of course does not preclude the existence of phenomena such as loss of stability in different environments, architectures, or training setups. More discussion on this can be found in Appendix B.

Figure 7: Mean average regret ($\downarrow$) on the hard suite. Lower regret corresponds to faster, more stable training. MAD-TD outperforms BRO.

**Continued training:** To highlight the pitfalls of resets, we employ a common RL theory metric the per timestep average regret

$$\overline{\text{Reg}}(T) = \frac{1}{T} \sum_{t=0}^{T-1} (\mathcal{R}^* - \mathcal{R}_t) \quad,$$

where $\mathcal{R}_t$ denotes the approximate cumulative return in episode $t$ and $\mathcal{R}^*$ the optimal return. We use the maximum return any of the algorithms achieved $\hat{\mathcal{R}}^*$ as a lower bound on the optimal return $\mathcal{R}^*$. Regret quantifies how much better the algorithm could have performed throughout training. In other words, in situations where continued learning is crucial, such as many safety critical applications, regret might be a better measure of performance. It captures not only how good the final policy is, but also how well the algorithm adapts over time, and minimizes mistakes. We present a comparison

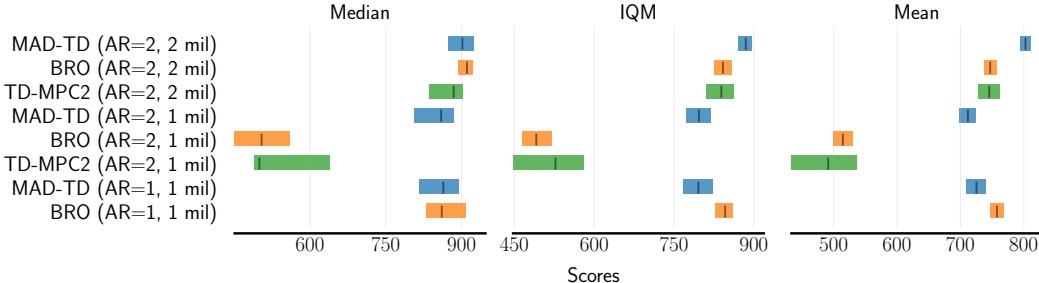

Figure 5: Performance comparison on the hard tasks for MAD-TD, BRO, and TD-MPC, with varying number of steps and action repeat settings. MAD-TD is on par with all baselines, has higher mean and IQM when trained for 2 million time steps and action repeat 2, and strongly outperforms TD-MPC2 and BRO at 1 million time steps with action repeat 2.

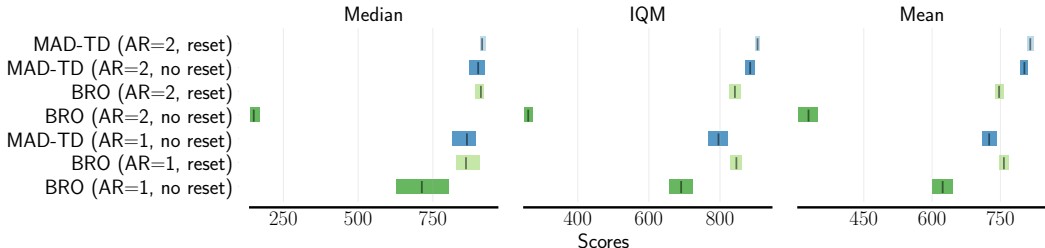

Figure 6: Resetting evaluation of MAD-TD and BRO. Lighter color denotes performance with reset, and darker without. While MAD-TD's performance only increases slightly when adding resetting, BRO is unable to achieve strong performance in and setting without resetting.

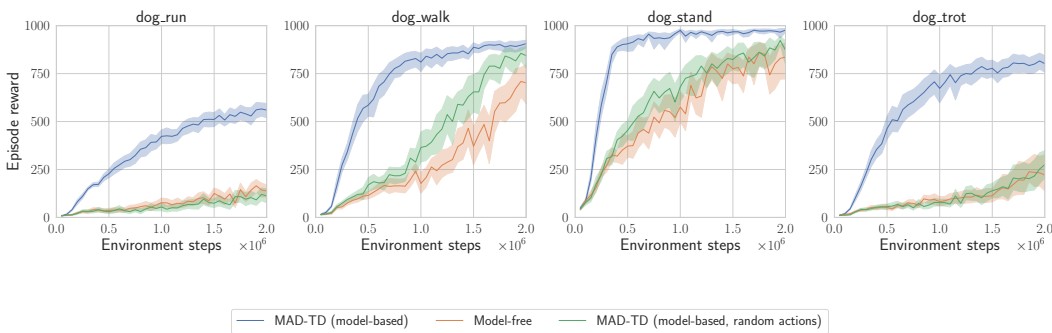

Figure 8: Return curves for the dog tasks when using on-policy, random and no model-generated data. When generating model-based data with random actions, performance of MAD-TD drops close to the model-free baseline, highlighting the importance of *on-policy* actions.

of MAD-TD and the resetting-based BRO in Figure 7 using an action repeat of 1. The results show, even though both algorithms are close in their final return, their training behavior differs vastly. MAD-TD has lower regret showcasing its strength in continued deployment.

## 5.3 FURTHER EXPERIMENTS AND ABLATIONS

To further test our approach, we present two additional experiments on the *hard suite*: changing the action selection for the model data generation, and reducing the model performance. In addition, we investigate the impact of using model based data on the smoothness of the learned value function.

**Off-policy action selection in the model:** To verify that the improvement in performance is due to the off-policy correction provided by the model, we repeat the *hard suite* experiments with a UTD of 8 and 5% model data, but we chose actions randomly from a uniform distribution across the action space. The results are presented in Figure 8. They highlight that random state-action pairs do not provide the necessary correction and the performance deteriorates to that of the model-free baseline.

**Smaller model networks:** To study the effect of the modeling capacity on our method, we ablate the size of the latent model by reducing the network size across the hard suite. The results are presented in Figure 9. We see that reducing the network size has an immediate and monotonic impact on the performance of our approach, suggesting that the model learning accuracy and prediction capacity is indeed vital for our approach to function well. However, even with small models of 64 hidden units, we still see some benefits from training with the model predicted data.

**Perturbation robustness of the model-corrected values:** To motivate our method, we conjectured that one problem with training actor-critic methods is that the actor conducts a quasi adversarial search for overestimated values on the learned critic.[1] To substantiate this claim, we used the iterated projected gradient method Madry et al. (2018) to estimate the smoothness of the learned value

---

[1]*Quasi* because the actor is not constrained to find an action close to the replay buffer sample.

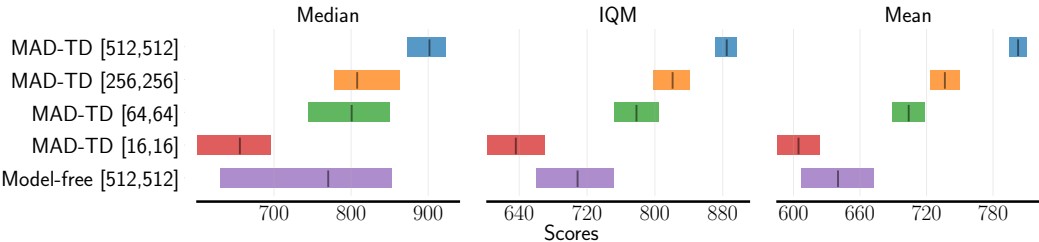

Figure 9: Performance evaluation when reducing the model size of the latent model in MAD-TD. The performance predictably drops with decreasing hidden layer size, however only strongly decreasing the model size below 64 reduces the performance below that of the model-free ablation.

functions on the humanoid environments at a UTD of 1 with and without model data. The results in Figure 10 show that not using any model data leads to value functions with higher oscillations, either across the whole training run (humanoid_run), or in the middle of training (stand and walk).

## 6    CONCLUSION

Our experiments allow us to conclude that wrong generalization of the value functions to unseen, on-policy actions is indeed a major challenge that prevents stable off-policy RL, both in theory and in practice. Model-Augmented Data for Temporal Difference learning (MAD-TD) is able to leverage the learning abilities of latent self-prediction models to provide small, yet crucial amounts of on-policy transitions which help stabilize learning across the hardest DeepMind Control suite tasks. With a relatively simple model architecture and learning algorithm, this method proves to be on par with, or even outperform other strong approaches, and does not rely on mechanisms such as value function ensembles or resetting which were previously conjectured to be necessary for stable learning in high UTD regimes. However, we highlight limitations of the approach in Appendix A.

Our work opens up exciting avenues for future work. The issue of poor generalization in off-policy learning can likely be tackled with other approaches such as diffusion models (Lu et al., 2024) or better pre-trained foundation models, and our presented experiments provide an important baseline for such work. Furthermore, while we have purposefully kept our approach as simple as possible to validate our hypothesis, many ideas from the model-based RL community such as uncertainty quantification (Chua et al., 2018; Talvitie et al., 2024), multi-step corrections (Buckman et al., 2018; Hafner et al., 2020), or policy gradient estimation (Amos et al., 2021) can be combined with our approach. Our insight that surprisingly little data is necessary to achieve strong correction can likely be leveraged in these other approaches as well to trade-off model errors and value function errors more carefully. Finally, while we chose the data to roll out in our models at random, our insights can likely be combined with ideas from the area of DYNA search control (Pan et al., 2019; 2020) to select datapoints on which the correction has the most impact.

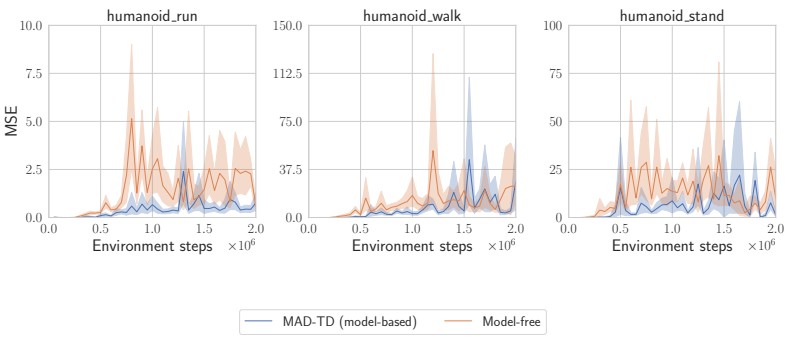

Figure 10: Magnitude of the difference between $Q(x, \pi(x))$ and $Q(x, \tilde{a})$, where $\tilde{a}$ is an adversarial perturbation of $\pi(x)$. We see larger perturbation for the runs without model correction data.

ACKNOWLEDGMENTS

We thank the members of the TISL and AdAge labs at the University of Toronto for enlightening discussions. We acknowledge the great help of Evgenii Opryshko, Taylor Killian, Amin Raksha, Maria Attarian, and Heiko Carrasco for providing detail feedback on our writing, and Evgenii for help with the experiments. For the adversarial experiments, we received helpful advice from Avery Ma, Jonas Guan, and Anvith Thudi.

We thank the anonymous reviewers for their helpful feedback and in-depth discussion.

EE and MH's research was partially supported by the Army Research Office under MURI award W911NF20-1-0080, the DARPA Triage Challenge under award HR00112420305, and by the University of Pennsylvania ASSET center. Any opinions, findings, and conclusion or recommendations expressed in this material are those of the authors and do not necessarily reflect the view of DARPA, the Army, or the US government.

AMF acknowledges the funding from the Natural Sciences and Engineering Research Council of Canada (NSERC) through the Discovery Grant program (2021-03701). CV acknowledges the funding from the Ontario Graduate Scholarship. Resources used in preparing this research were provided, in part, by the Province of Ontario, the Government of Canada through CIFAR, and companies sponsoring the Vector Institute.

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

# A    LIMITATIONS

The core limitation of our methodology relies in the assumption that a sufficiently strong environment model can indeed be learned online. While a proof of feasibility exists for many interesting RL benchmarks in the forms of the Dreamer (Hafner et al., 2021) and TD-MPC2 (Hansen et al., 2024) lines of work among many others, for a completely novel environment a practitioner will still have to test if current model learning schemes are sufficient to achieve strong control policies.

Furthermore, we can only generate data from the states visited under a past policy. There is still a difference between the state distribution of the replay buffer, and the target policy stationary distribution. While this difference does not seem to lead to catastrophic failures in the DMC benchmarks, the distribution shift might be more problematic in other environments.

Finally, we observe an interesting failure cases of our idea: in some simple environments we surprisingly observe worse performance with our network architectures compared to the BRO baseline. This issue is likely due to the fact that the TD-MPC2 architecture is tuned for learning in complex high-dimensional problems, which leaves it potentially over-parameterized on simple tasks.

While our work shows that reduced learning capacity due to plasticity does not seem to be the major contributor to learning problems in benchmarks like DMC, that does not exclude the possibility that related issues appear nonetheless after accounting for the off-policy value estimation problem. We did not test increasing the reset ratio even further as other prior work has done, as we already observed no benefits from increasing the replay ratio from 8 to 16 in most of our experiments and performed on par or beyond previous baselines. Issues in reinforcement learning are often entangled in a complex way, e.g. a failure in exploration can lead to stagnant data in the replay buffer which prevents a critic from further improving its estimates, leading to worse exploration and so on.

# B    EXTENDED RELATED WORK

Beyond mitigating value function overestimation and unstable learning (see Subsection 3.3), other works have approached the difficulty of off-policy learning and high update ratios from other perspectives. Here, we survey further related papers which do not provide direct background for this work, but are nonetheless relevant as either alternative approaches or possible enhancements.

**Other ways of incorporating off-policy data**    Having access to more diverse data has been shown to be beneficial for reinforcement learning, when this data is carefully used to mitigate the problems resulting from off-policy training. Ball et al. (2023) show that a large offline replay buffer can be used to improve training by sampling online training batches both from online data and offline data, and labelling the offline transitions with a reward of 0. Agarwal et al. (2022) and Tirumala et al. (2024) also highlight that previously collected replay buffers can be used to improve training performance on agents. In this work, we focus on the online setting where we do not have access to a replay buffer of previously collected transitions. These ideas however can easily be combined by e.g. training a model from an available larger offline data buffer.

**SynthER**    Another related approach to obtain additional data is the diffusion-based method proposed by (Lu et al., 2024). In this work, the replay buffer data is augmented with additional samples obtained from a diffusion model that is trained on the replay buffer. The underlying hypothesis of SynthER is that the failure of high-UTD learning stems mostly from a lack of diverse data in the replay buffer. They demonstrate on the easier DM Control tasks that simply adding data from a generative model can be beneficial to learning. This is opposed to our hypothesis, which claims that high-UTD learning is difficult specifically due to the lack of off-policy action corrections. As SynthER does not provide results on the hard DMC tasks, we reran the original code to compare our claims The results and a discussion can be found in Subsection E.4.

In the online off-policy regime, Fujimoto et al. (2024) recently proposed TD7, which incorporates similar architectural choices to MAD-TD. They use a self-predictive encoder to learn good state representations, but concatenate them with the state and action representation provided by the environment to limit loss of information. This design choice proved to be beneficial but would require learning a observation-space next-state prediction, which is difficult in practice, especially in high dimensional environments. To address the policy distribution shift, TD7 does not update the actor

at every timestep but instead collects several full trajectories with a fixed policy and then conducts update steps afterwards. However, this interval still needs to be balanced as a hyperparameter. TD7 was not evaluated on DMC, which is why we do not present a comparison.

**Model-based reinforcement learning**   As surveying model-based reinforcement learning is a rather sizable tasks, we refer readers to the survey by (Moerland et al., 2023) for reference. Decision-aware latent models such as the one Hansen et al. (2024) and we use have been studied specifically in several different variants. Silver et al. (2017) proposes a latent model that is trained with TD learning, which provides the basis for the Schrittwieser et al. (2020) algorithm. The addition of a latent self-prediction loss was first proposed by Li et al. (2023) to stabilize learning problems with the TD learning loss. This interplay was further studied by Ni et al. (2024) and Voelcker et al. (2024) in recent works.

From a theoretical angle, decision-aware losses similar to those used in MuZero where first studied by Farahmand et al. (2017) and Farahmand (2018). Grimm et al. (2020) and Grimm et al. (2021) further study the loss landscape and minimizers of such losses, while Kastner et al. (2023) studied the extension of the loss to distributional settings.

While previous works have called the stability of the VAML loss into question (Lovatto et al., 2020; Voelcker et al., 2022), we find that it is stable and performant when combined with the HL-Gauss representation Farebrother et al. (2024) and an auxiliary BYOL style loss Grill et al. (2020); Li et al. (2023). Compared to MuZero it is also significantly easier to implement.

A more thorough overview on the topic of decision-aware learning can be found by Wei et al. (2024).

**Offline reinforcement learning**   In the context of batch reinforcement learning or offline RL (Lange et al., 2012; Fujimoto et al., 2019), the action distribution shift is a known phenomenon. The main counter to the problem however does not rely on closing the generalization gap, but on explicit pessimistic regularization Jin et al. (2021). Such pessimistic regularization has been shown to be highly detrimental in online RL, as it removes the capability for the agent to explore its environment efficiently (D'Oro et al., 2023; Hussing et al., 2024). In offline RL, authors have explored the capability of models to provide some improvements to generalization (Yu et al., 2020). However, in online RL the community has mostly relied on the hope that additional optimistic exploration based on the value function will close the generalization gap without explicit interventions. We show that this is not the case.

**Loss of plasticity**   A phenomenon that was originally reported in continual learning is that tendency for neural network based agents to lose their ability to learn over time. This phenomenon has also been investigated in the realms of RL (Igl et al., 2021), as RL can effectively be thought of as a type of continual learning problem. Sometimes the phenomenon is referred to as plasticity loss (Lyle et al., 2021; Abbas et al., 2023). As highlighted before, we do not find strong evidence for the primacy bias or loss of plasticity during our experiments on the DMC suite.

However, that does not imply that the phenomenon does not exist. In fact, we believe that resolving stability issues such as those presented in our paper will help us to better isolate other nuanced issues such as plasticity loss more clearly. Previous studies have identified and combated plasticity loss using feature rank maximization (Kumar et al., 2021), regularization (Lyle et al., 2023), additional neural network copies (Nikishin et al., 2024), minimizing dormant neurons (Sokar et al., 2023; Xu et al., 2024), various neural network architecture changes (Lee et al., 2023), slow and fast network updates (Lee et al., 2024) or weight clipping (Elsayed et al., 2024).

It is unclear how many improvements obtained by these changes can be explained by divergence effects (Hussing et al., 2024) or stability issues such as those established in our work as there seems to be a non-zero overlap in techniques that combat either. Nauman et al. (2024a) have argued that many RL training problems can be difficult to disentangle from the plasticity loss phenomenon. An interesting direction of future work is to test for plasticity loss with well regularized off-policy value function learning, for instance by combining our method with separate solutions established for plasticity loss such as those from Lyle et al. (2024).

It is also not unlikely that the training dynamics of the state-based dense-reward tasks on the DMC suite are more benign than those found in Atari games. Many works on plasticity loss have stud-

ied sparse image-based control tasks with pure Q learning approaches, such as DQN on the Atari benchmark (Sokar et al., 2023; Lee et al., 2024). The problem may be more prevalent when replay buffers cannot be maintained in full and the RL setting becomes a true continual learning problem.

**Other stability perspectives** Our work studies the stability of losses during training. We highlight that forgoing resetting decreases regret as the executed policies are more stable in the sense that they are not reset at regular intervals. We also highlight that model-generated data can somewhat improve the stability of policies against adversarial attacks. However, there are other notions of *stability* that should be considered relevant and that are orthogonal to our work. Here we will give a non extensive overview into the different directions that exist as a starting point for the reader. For instance from a theoretical perspective, stability can be formulated as differential privacy (Vietri et al., 2020) or algorithmic replicability to obtain identical policies (Eaton et al., 2023). From a theoretical as well as practical perspective, issues such as robustness to adversarial attacks (Nilim & Ghaoui, 2005; Iyengar, 2005; Wiesemann et al., 2013; Pinto et al., 2017). Finally, from an empirical perspective robustness to hyperparameters (Ceron et al., 2024; Patterson et al., 2024) and attempts at variance reduction to get more reliable solutions (Anschel et al., 2017; Kuang et al., 2023) can be considered notions of stability.

## C   MATHEMATICAL DERIVATIONS

While the proof by Sutton (1988) which we use as a basis discusses the stationary distribution of the Markov chain $P^\pi$, we define our loss in terms of a discounted state-action occupancy. We therefore briefly prove an auxiliary result to extend the analysis to the case of discounted state occupancy probabilities. Note that when we talk about positive-definiteness, we use a definition which applies to potentially non-symmetric matrices, and merely requires that $u^\top X u\, 0$ for all vectors $u$.

**Proposition 1.** *Let $P$ be a stochastic matrix. Define the discounted state occupancy distribution $\mu$ of $P$ for some starting state distribution $\rho$ and some discount factor $\gamma \in [0, 1)$ as*

$$\mu^\top = (1 - \gamma) \sum_{n=0}^{\infty} \gamma^n \rho^\top P^n.$$

*Let $D$ be a diagonal matrix whose entries correspond to the discounted state occupancy distribution. Then the matrix $D(I - \gamma P)$ is positive definite.*

*Proof.* First, note that
$$(1 - \gamma)\rho^\top + \gamma\mu^\top P = \mu^\top$$
by the definition of $\mu$ and the properties of the infinite sum. Therefore,

$$\mu^\top P = \frac{1}{\gamma} \left( \mu - (1 - \gamma)\rho \right) \ .$$

Sutton (1988) asserts that a matrix $A$ is positive definite iff $A + A^\top$ is positive definite. Furthermore, if the diagonal entries of a symmetric matrix are positive and its off-diagonal entries are negative, then it suffices to show that the row and column sums of matrix are positive.

For
$$D(I - \gamma P) + (I - \gamma P^\top)D^\top$$
the off-diagonal terms are clearly non positive as D is diagonal. On the main diagonal, we have $2(\mu_i - \gamma p(i|i)\mu_i)$ which is positive as $p(\mu_i|\mu_i) \leq 1$. It now suffices to show that the row and column sums of $D(I - \gamma P)$ are positive. For the row sum, we can make use of the fact that $P$ is a stochastic matrix, so
$$D(I - \gamma P)\mathbf{1} = D(\mathbf{1} - \gamma\mathbf{1}) \geq 1 \ .$$

For the column sum, we make use of the fact that $\mathbf{1}D = \mu$. Then

$$\mu(I - \gamma P) = \mu - \gamma\frac{1}{\gamma}\left(\mu - (1 - \gamma)\rho\right) = (1 - \gamma)\rho \geq 1 \ .$$

As $\rho$ is a probability vector the final inequality holds for all $\gamma \in [0, 1)$.

All conditions presented by Sutton (1988) hold, and therefore we have $D(I - \gamma P)$ is positive definite. $\square$

To derive the gradient flow stability conditions in Subsection 3.1, we first restate the loss function

$$L(\theta) = \sum_{i=1}^{n} \left[ D^{\pi_i} \left( \Phi^\top \theta - [R + \gamma P^\pi \Phi^\top \theta]_{\text{sg}} \right)^2 \right] . \tag{7}$$

The stability of learning with this loss can be analyzed using the gradient flow (Sutton et al., 2016). To derive the gradient flow, we compute the gradient of the loss function with regard to the parameters $\theta$. As the loss has a relatively simple quadratic form and the derivative is a linear transformation, it decomposes nicely as

$$\nabla_\theta L(\theta) = 2\Phi \sum_{i=1}^{n} D^{\pi_i} \left( \Phi^\top \theta - R - \gamma P \Pi \Phi^\top \theta \right) \tag{8}$$

$$= 2\Phi \sum_{i=1}^{n} D^{\pi_i} \left( (I - \gamma P \Pi) \Phi^\top \theta - R \right) \tag{9}$$

$$= 2\Phi \sum_{i=1}^{n} D^{\pi_i} (I - \gamma P^\pi) \Phi^\top \theta - 2\Phi \sum_{i=1}^{n} D^{\pi_i} R . \tag{10}$$

Using the equation for the gradient flow $\dot{\theta} = -\frac{\eta}{2} \nabla_\theta L(\theta)$ with learning rate $\frac{\eta}{2}$, we obtain

$$\dot{\theta} = -\eta \Phi \sum_{i=1}^{n} D^{\pi_i} (I - \gamma P^\pi) \Phi^\top \theta + \eta \Phi \sum_{i=1}^{n} D^{\pi_i} R , \tag{11}$$

This gradient flow is guaranteed to be stables (meaning it will not diverge around the stationary point $\theta^*$) if the key matrix $\sum_{i=1}^{n} D^{\pi_i} (I - \gamma P \Pi)$ is positive definite (Sutton, 1988).

We can decompose our key matrix into the on-policy key matrix and a remainder easily

$$\sum_{i=1}^{n} D^{\pi_i} (I - \gamma P \Pi) \tag{12}$$

$$= \sum_{i=1}^{n} D^{\pi_i} (I - \gamma P \Pi + \gamma P \Pi_i - \gamma P \Pi_i) \tag{13}$$

$$= \sum_{i=1}^{n} D^{\pi_i} (I - \gamma P \Pi_i) + \gamma \sum_{i=1}^{n} D^{\pi_i} P (\Pi_i - \Pi) . \tag{14}$$

The first group of summands are all positive definite, following Proposition 1. As the sum of positive definite matrices is positive definite, the claim stands.

However, the second group has no such guarantees. This highlights the role that the target policy action selection plays in the stability of Q learning.

## D    IMPLEMENTATION

Our experiments are implemented in the jax library to allow for easy parallelization of multiple experiments across seeds. All networks follow the standard architecture from Hansen et al. (2024) with two changes: instead of using an ensemble of critics, we opt for a single double critic pair. We also do not use a stochastic policy, instead simply using a deterministic network with a tanh activation as used in Lillicrap et al. (2016); Fujimoto et al. (2018). Full hyperparameters are presented in Table 2 and the architecture can be found in Table 1. We use mish activation functions (Misra, 2020) and the Adam optimizer to train our models (Kingma & Ba, 2015). For reference, our code is available at https://github.com/adaptive-agents-lab/mad-td.

| Encoder $\Phi$ | Dense Layer | Mish | in_size=$|\mathcal{X}|$, out_size=512 |
|---|---|---|---|
| | Dense Layer | Simnorm(8) | out_size=512 |
| Latent Model $F$ | Dense Layer | Mish | in_size=512 + $|\mathcal{A}|$, out_size=512 |
| | Dense Layer | Mish | out_size=512 |
| | Dense Layer | Simnorm(8) | out_size=512 |
| Q head $\hat{Q}$ | Dense Layer | Mish | in_size=512 + $|\mathcal{A}|$, out_size=512 |
| | Dense Layer | Mish | out_size=512 |
| | Dense Layer | – | out_size=1 |
| Actor $\hat{\pi}$ | Dense Layer | Mish | in_size=512, out_size=512 |
| | Dense Layer | Mish | out_size=512 |
| | Dense Layer | tanh | out_size=$|\mathcal{A}|$ |

Table 1: Network architecture for MAD-TD.

| Parameter | | Parameter | |
|---|---|---|---|
| Initial steps (Random policy) | 5000 | HL-Gauss vmax | $150 \cdot \mathrm{AR}$ |
| Batch size | 512 | HL-Gauss num bins | 151 |
| RL learning rate | 0.0003 | Model data proportion | 0.95 |
| Model learning rate | 0.0003 | Reset interval (where applicable) | 200000 |
| Encoder learning rate | 0.0001 | Model & encoder update ratio | 1 |
| Soft update $\tau$ | 0.995 | Actor & critic update ratio | varying |
| Discount factor $\gamma$ | 0.99 | MPC number of samples | 512 |
| Model forward prediction steps | 1 (4 for AR=1) | MPC iterations | 6 |
| Gradient clipping | 10.0 | MPC top k | 64 |
| HL-Gauss vmin | $-150 \cdot \mathrm{AR}$ | MPC temperature | 0.5 |

Table 2: Hyperparameters. We adapted three parameters to the action repeat = 1 setting, as the magnitude of the reward changes.

**Loss functions:** As we use the HL-Gauss representation (Farebrother et al., 2024) for the critic, the loss is the cross-entropy between the estimated Q function's categorical representation $Q_{\mathrm{rep}}$ and the bootstrapped TD estimate,

$$\mathcal{L}_Q = \sum_{i=1}^{m} \mathrm{TD}(\hat{Q}_{\mathrm{rep}})_i \log \hat{Q}_{\mathrm{rep}_i} \;,$$

where the indices $i$ denote the positions of the categorical vector representation used by HL-Gauss. This is the same loss that is used for the two-hot encoding in Hansen et al. (2024), the only difference is the target encoding function. For more details, see Farebrother et al. (2024).

We use a latent encoder $\phi : \mathcal{X} \times \mathcal{A} \to \mathcal{Z}$ that maps into the SimNorm space, the space of n k-dimensional simplices (Lavoie et al., 2023). Writing $\hat{p}$ for the learned world model and $\hat{r}, \hat{x}' \sim \hat{p}(\cdot|x, a)$ for reward and next latent-state samples, the loss for our model and encoder is

$$\mathcal{L}_{\mathrm{model}}(x, a, r, x') = \mathcal{L}_{\mathrm{rew}}(x, a, r) + \mathcal{L}_{\mathrm{forward}}(x, a, x') + \mathcal{L}_Q(x, a, r, x') \tag{15}$$

$$\mathcal{L}_{\mathrm{rew}}(x, a, r) = (r - \hat{r})^2 \tag{16}$$

$$\mathcal{L}_{\mathrm{forward}} = -\sum_{i=1}^{n \cdot k} \phi(x')_i \log \hat{x}'_i \;, \tag{17}$$

where the index $i$ is again element-wise across the simplex representation used for the latent state. Note that we propagate the critic learning gradients into the encoder only for the real data and not the model generated one to prevent instability.

**Baseline results** We took available results from Nauman et al. (2024b) and Hansen et al. (2024) for all plots where possible, and used the official implementation of BRO to rerun the experiments without resetting and with differing action repeats. Other hyperparameters were left as-is.

# E FURTHER RESULTS

## E.1 Q VALUE OVERESTIMATION

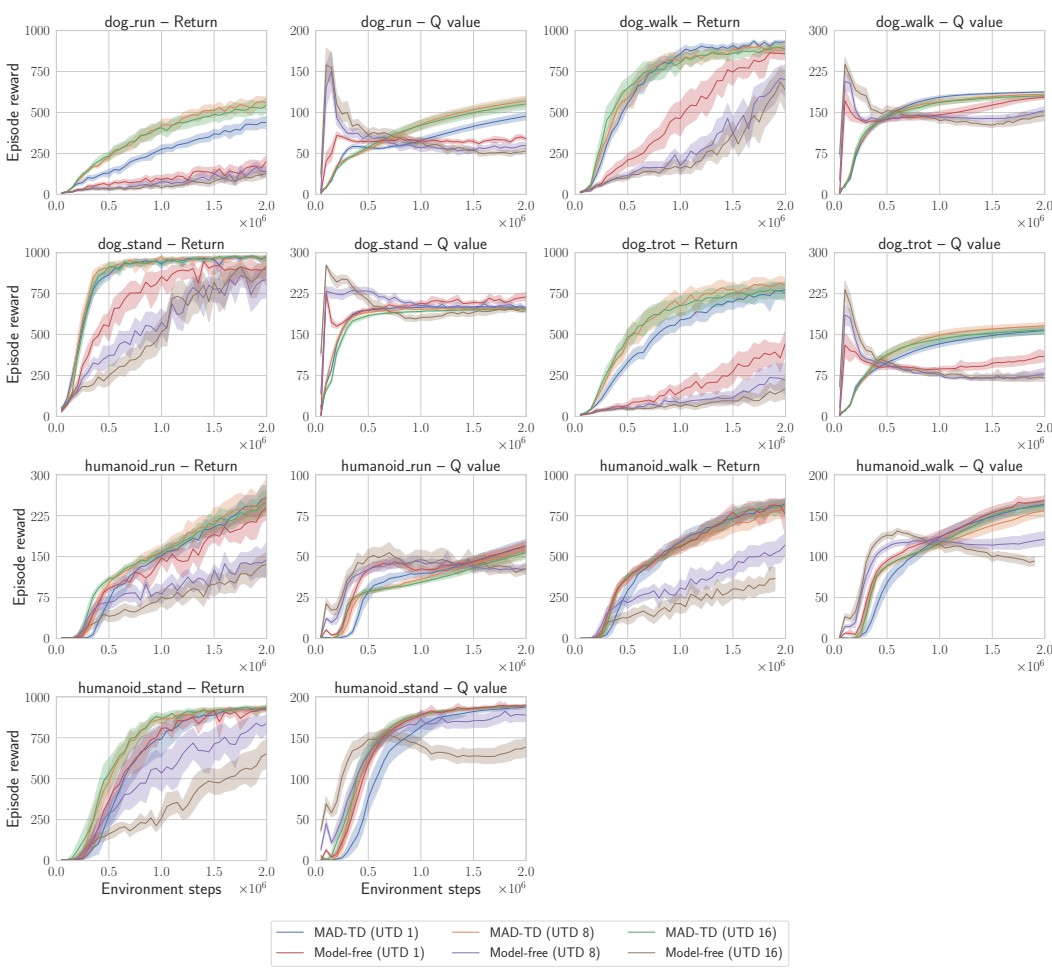

Figure 11: Return curves and Q values with differing UTD values.

We plot the return curves and corresponding Q estimates for different UTD values and with and without model-generated data on the hard suite. The results are presented in Figure 11. As we see, across all tasks the model free variant strongly overestimates the Q values, especially in the beginning.

## E.2 Humanoid Results

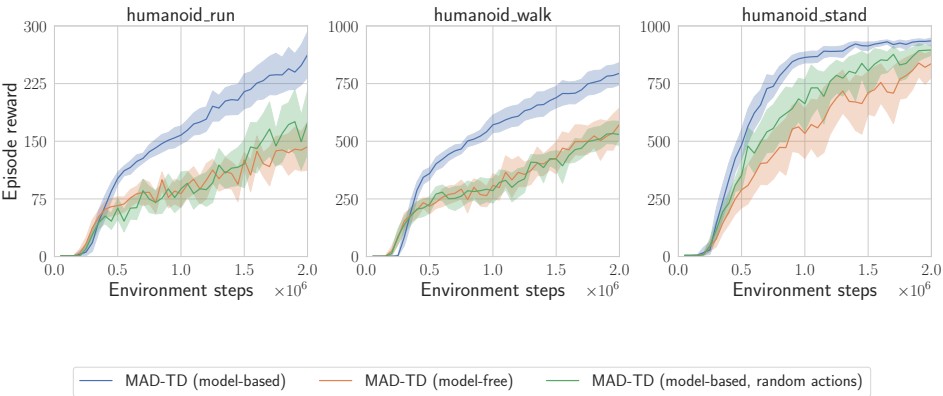

Figure 12: Return curves for the humanoid tasks when using on-policy (blue), random (green) and no model-generated data (orange). The observed performance impacts are comparable to the dog case.

For several experiments, we only showed the dog results from the main suite to avoid cluttering the main body of the paper. The corresponding humanoid results are presented in Figure 11 and Figure 12, corresponding to Figure 3 and Figure 8 respectively. As the plots highlight, the main insights transfer across the hard tasks.

### E.3 DIFFERENT QUANTITIES OF MODEL DATA

We evaluate using more model data to update our value functions and provide the results in Figure 13. Aggregated scores are presented in Figure 14. We observe that the majority of gain is obtained when using limited amount of model data, and larger amounts only provided limited gains in some humanoid runs. When using high amounts of model generated data, we observe deteriorating performance, which implies that the agent learns to exploit the model instead of solving the real task. This observation is consistent with similar observation about model exploration in prior work Zhao et al. (2023).

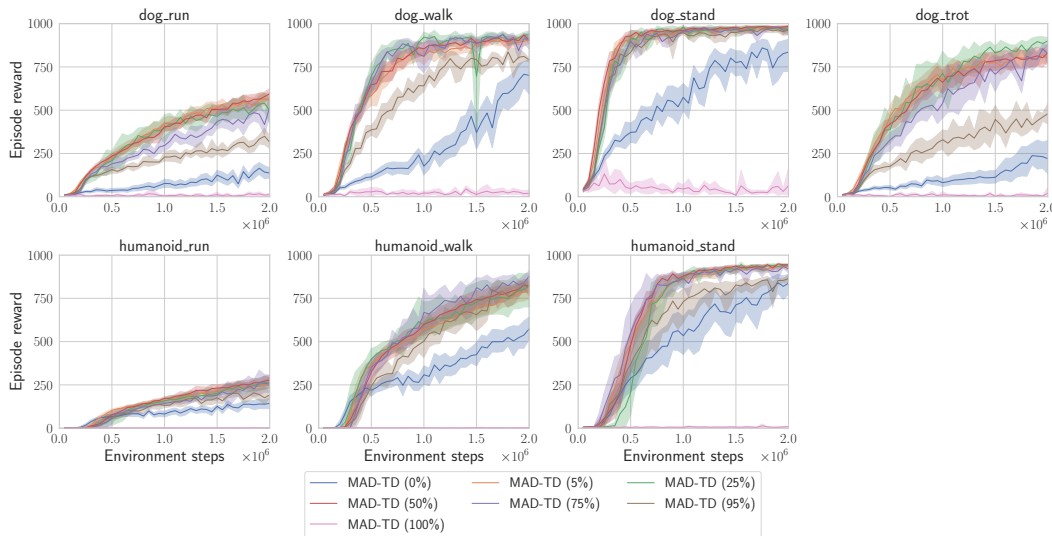

Figure 13: Return curves on the hard suite. We see that using substantially more data than 5% does not improve performance in a statistically significant way.

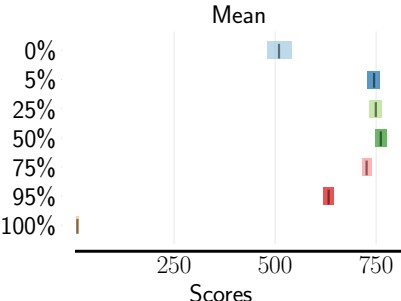

Figure 14: Aggregate statistics for differing values of $\alpha$ (amount of model data used) at UTD 8.

### E.4 SYNTHER COMPARISON

We present a comparison of our method and SynthER on the hard DMC tasks. Results can be found in Figure 15.

As is evident from the lack of strong performance of SynthER, merely increasing the amount of generated data is insufficient to combat the failure of learning at high UTD. We find that the Q values of the SynthER agents quickly diverge on all tasks in which it is unable to learn. This strengthens our hypothesis that for hard tasks, off-policy action correction is vital to achieve strong results.

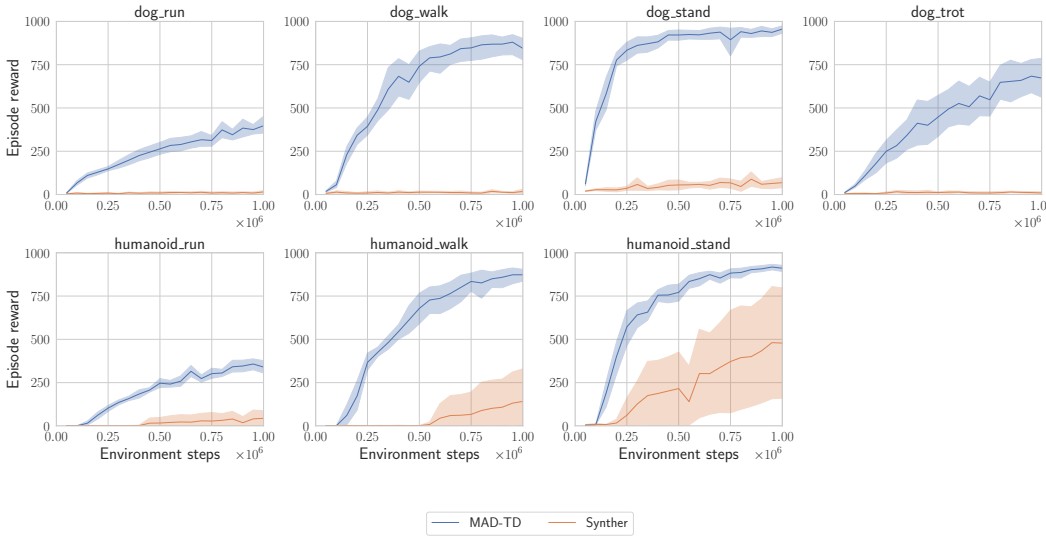

Figure 15: Performance curves for MAD-TD and SynthER on the hard DMC tasks. SynthER fails to achieve nontrivial results on most tasks, only outperforming a random policy on the humanoid walk and stand tasks.

## E.5 TD-MPC2 ABLATION

As described in the main paper, we simplify the base model of TD-MPC2 to improve the computational efficiency of the algorithm. This is necessary to conduct high UTD experiments. Here, we present a direct comparison of the original TD-MPC2 model, and our adapted version (Figure 16). We compare MAD-TD without any model generated data, at UTD 1, which corresponds to the standard setting of TD-MPC2. As pointed out in the main paper, all of our changes to the base model boil down to setting different hyperparameters, such as the rollout length, to achieve faster learning.

We find that this does not significantly change the overall results achieved by the base model, and we are therefore confident to attribute performance gains to our presented method.

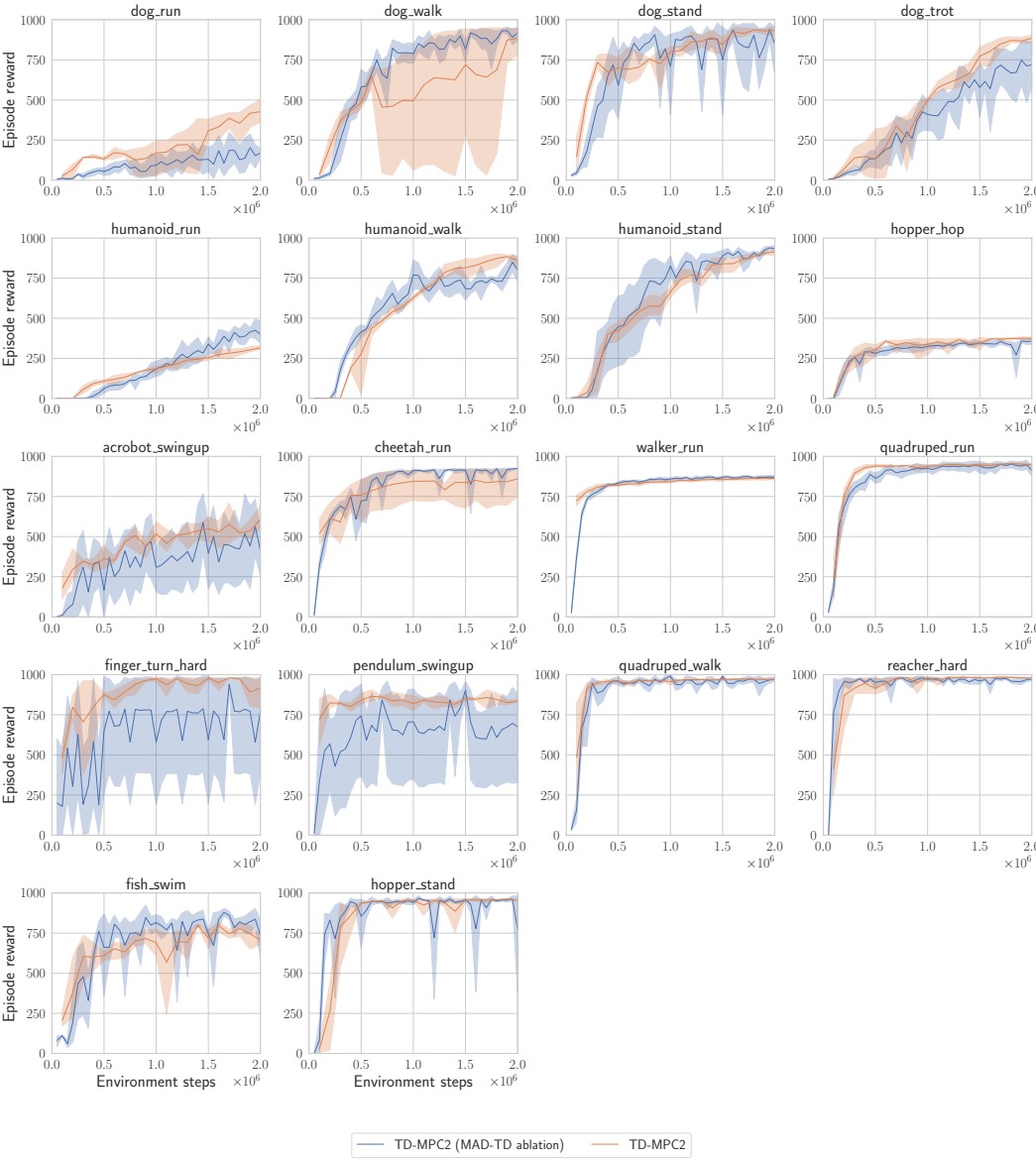

Figure 16: Performance variation of the base MAD-TD model compared to TD-MPC2. Our changes only very few times lead to lower performance which is acceptable given the large reduction in computational cost.

### E.6 MAD-TD, BRO, TD-MPC2 PER ENV ON THE HARD SUITE

We present the return curves for MAD-TD and the baselines per environment on the hard suite. Figure 17 shows the results with action repeat 2 and Figure 18 with action repeat 1. Perhaps surprisingly, the results of the algorithms are not fully consistent across this regime. Partially, this can be explained by the fact that our method and TD-MPC2 were first developed in the regime of action repeat 2, while BRO was only evaluated in the action repeat 1 setting. This suggests that the performance of each method depends in a non-trivial fashion on hyperparameter tuning. Yet, across both action repeat setting MAD-TD outperforms BRO without resetting consistently and only under-performs any previous algorithm on the dog trot task in the action repeat 1 setting.

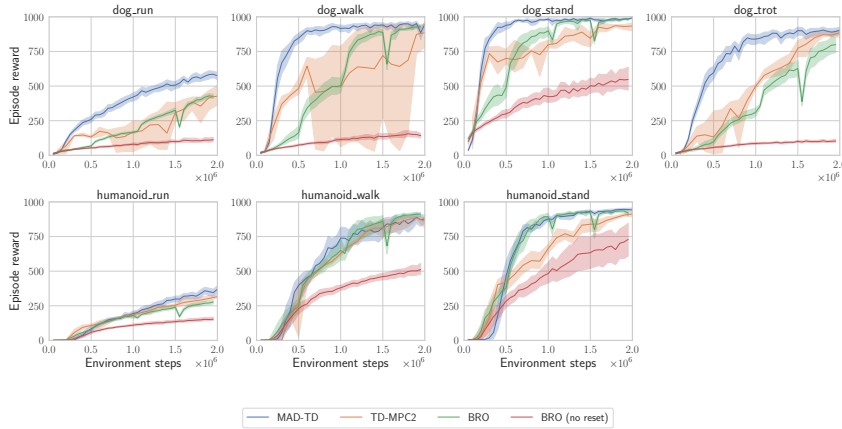

Figure 17: Return curves with action repeat set to 2.

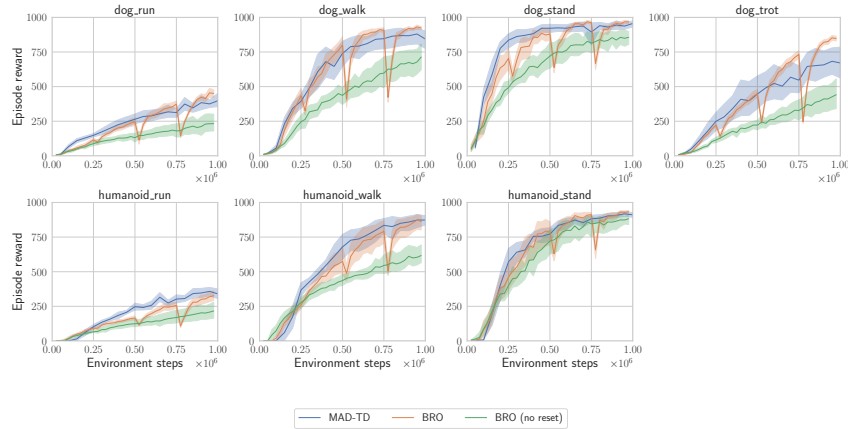

Figure 18: Return curves with action repeat set to 1.

We conjecture that the remaining gap in performance seems to be most likely attributable to exploration and optimism. While we focus on learning accurate value functions, Bro contains several components which are specifically designed to improve exploration. Investigating the tension between exploration and accurate value function fitting is an important direction for future work.

Bro and TD-MPC2 are explicitly evaluated without their exploration bonuses in separate evaluation rollouts. We however do not conduct such as separate evaluation as we do not add any additional exploration noise to our training. When plotting training performance, the gap between MAD-TD and Bro further closes, suggesting an important trade-off between test time and training performance.

## E.7 RESULTS ACROSS FURTHER DMC ENVIRONMENTS

We conducted more experiments on all DMC environments which were shown to benefit from the interventions in prior work (D'Oro et al., 2023; Nauman et al., 2024b).

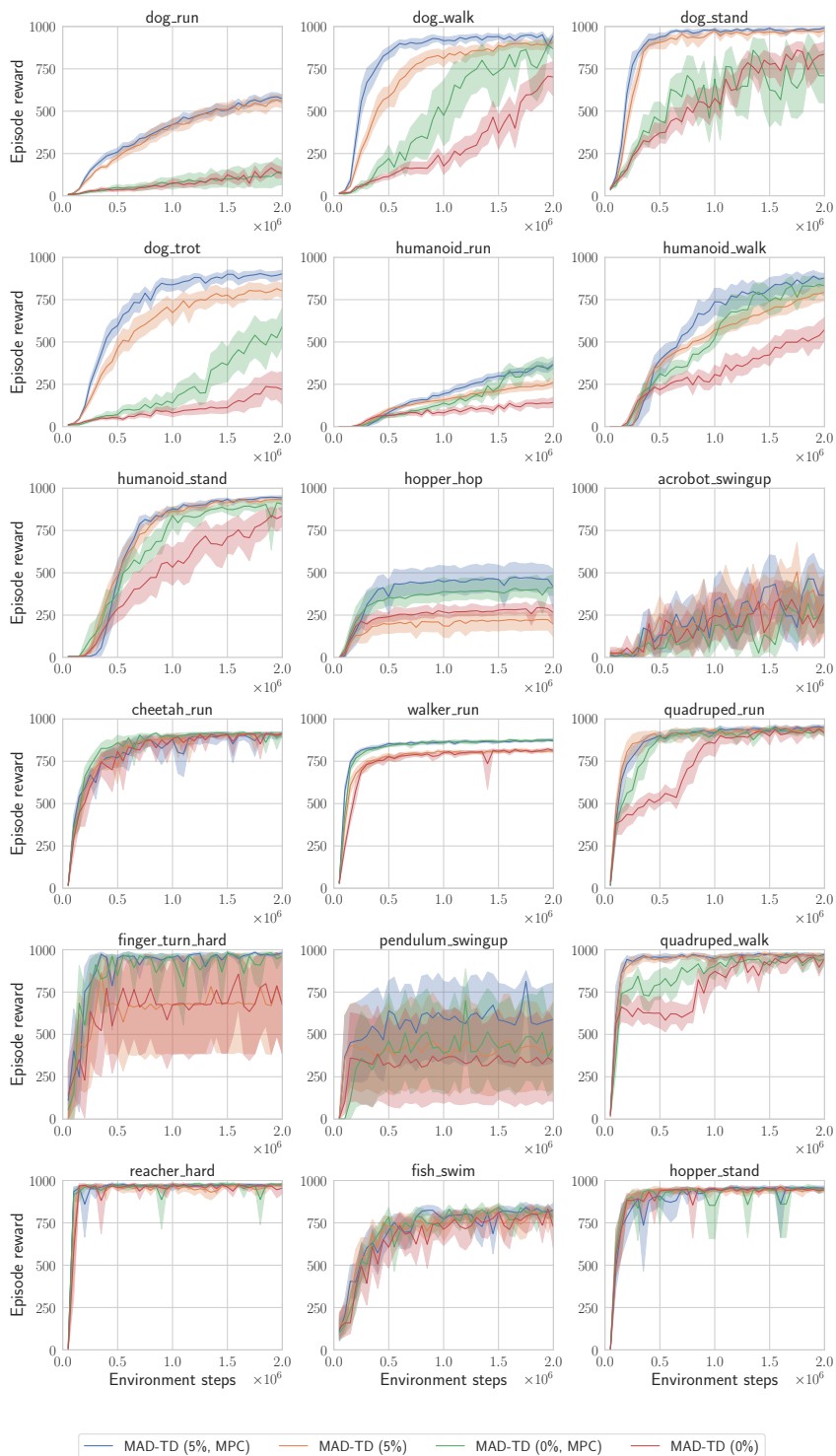

Figure 19: Return curves evaluating the impact of model-based data for critic learning and MPC. Overall, MPC and model-based critic learning both stabilize the learning process, as conjectured.

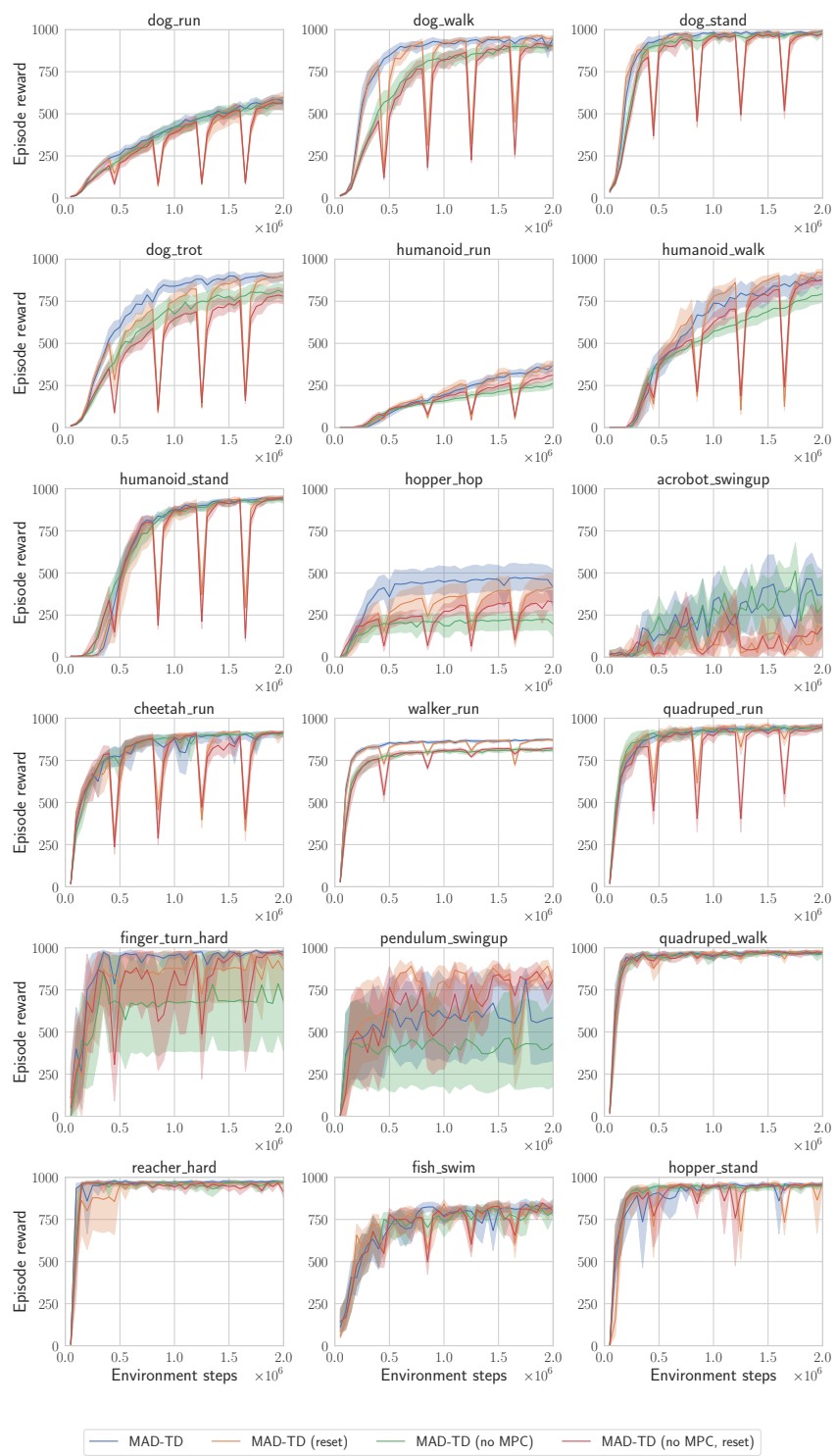

Figure 20: Return curves for the impact of resetting on MAD-TD with and without MPC. Without MPC, resetting can still improve the performance, but with MPC, we see no significant benefits from resetting across environments except pendulum. The hopper results highlight the importance (and danger) of the reset interval, as seemingly the reset algorithm is not able to recover "in time" to improve performance.

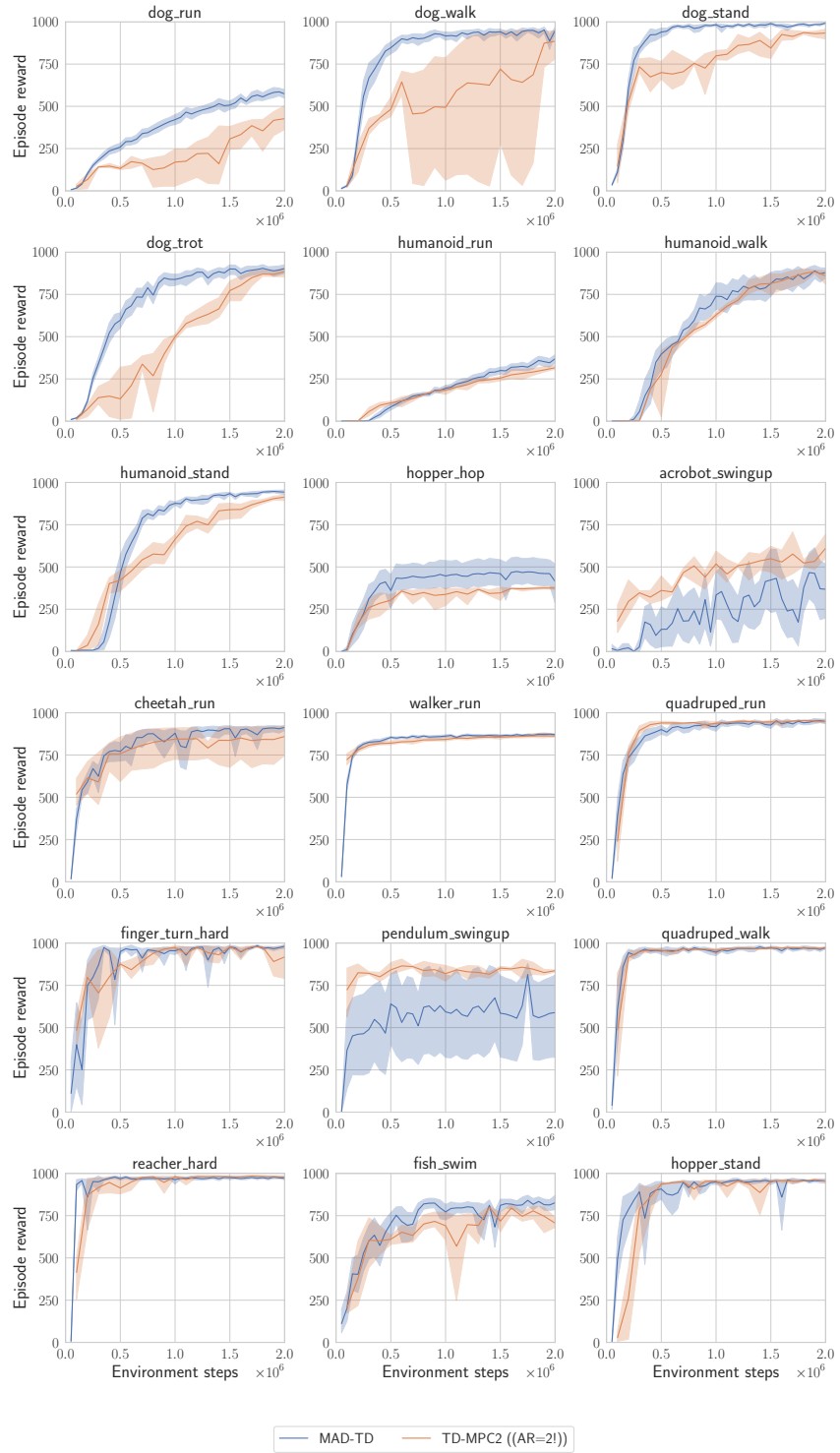

Figure 21: Comparison of MAD-TD and TD-MPC2 across more environments of the DMC suite. We observe gains compared to TD-MPC2 in the hard tasks, especially in terms of early learning performance, while TD-MPC2 has advantages on the pendulum_swingup and acrobot_swingup tasks. These seem to be exploration and stability issues for which the longer model rollouts of TD-MPC2 seem to help.

### E.8    METAWORLD

To broaden the basis of comparison, we compare our method to BRO and TD-MPC2 on 9 selected environments from the metaworld suite. Results can be found in Figure 22.

Overall, we observe that our method performs strongly on tasks in which the agent has access to a dense reward, such as *lever-pull* and *button press*. MAD-TD demonstrates the ability to quickly and stably bootstrap reward when available. When exploration is a challenge, learning can take longer with MAD-TD. Strong exploration for high-UTD algorithms is not the focus of MAD-TD and remains an open problem (Hussing et al., 2024). This is consistent with our core hypothesis: high UTD learning benefits in cases where fitting a correct value function is challenging. In tasks such as *pick-place-wall* the core challenge is exploration, as the agent receives no reward signal for the majority of early training. We therefore cannot expect high UTD learning to improve the performance in these tasks.

As pointed out, BRO and to a lesser extent TD-MPC2 have the benefit of exploring with optimism bonuses and ensembled value functions. We removed these from our method to cleanly study the impact of model generated data. However, improvements to exploration are mostly orthogonal to our proposed method and can be freely combined in future work.

Finally, as also shown by Nauman et al. (2024b), there is a curious failure case of TD3 compared to SAC in the case of environments with sparse rewards. In the absence of the entropy penalty form the SAC loss function, the tanh policy of TD3 tends to saturate, which can stymie exploration completely. This is, to the best of our knowledge, not discussed in the literature, and should be investigated in future work.

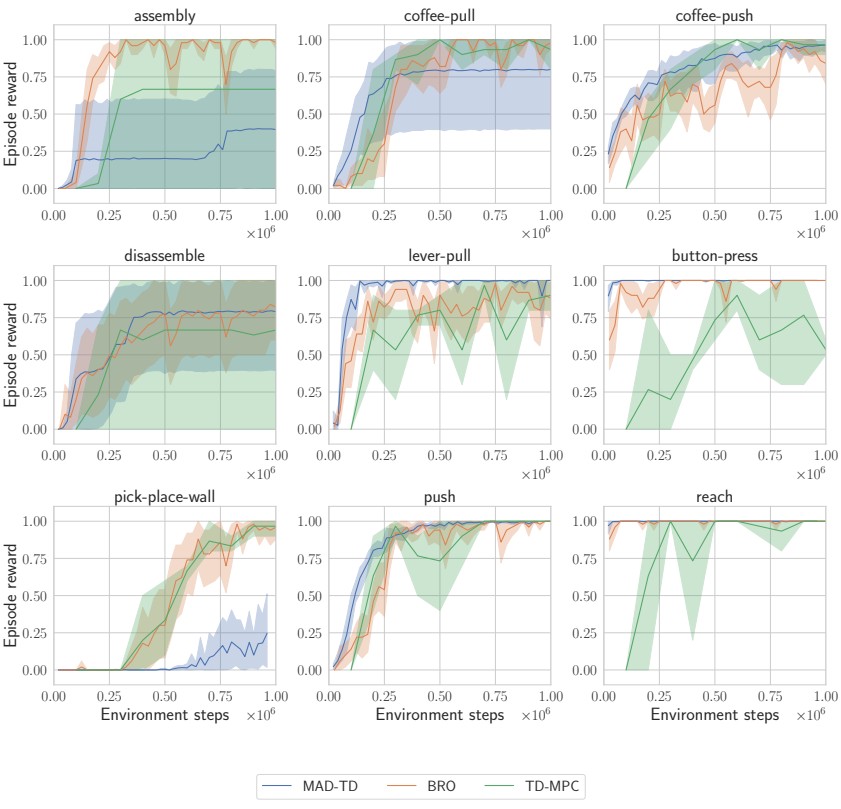

Figure 22: Performance comparison on Metaworld between MAD-TD, BRO, and TD-MPC2. MAD-TD performs strongly on tasks which provide sufficient reward information to bootstrap the value function quickly, while learning more slowly on sparse reward tasks. This is consistent with the core goal of our algorithm, to stabilize and improve value function learning.

