# OpenReview forum: "MAD-TD: Model-Augmented Data stabilizes High Update Ratio RL"
_ICLR.cc/2025/Conference — ICLR 2025 Spotlight_

### Official Review · Reviewer_UXUR · 2024-10-29

**Soundness:** 3
**Presentation:** 2
**Contribution:** 2
**Rating:** 6
**Confidence:** 3

**Summary:**

This paper mainly investigates the issue of overestimation of unseen on-policy actions and instability caused by a high update-to-data (UTD) ratio in off-policy reinforcement learning via theoretical analysis and experiments. To address this issue, the authors introduce a new method named Model-Augmented Data for Temporal Difference Learning (MAD-TD), which combines model-generated synthetic data with real data to enhance and stabilize the off-policy RL training process. The experiments conducted on the DeepMind Control benchmark demonstrate that MAD-TD outperforms other baselines and leads stable learning even in high UTD settings.

**Strengths:**

1. The authors provide thorough theoretical analysis and experimental results, demonstrating the effectiveness of MAD-TD in stabilizing the training process and achieving strong performance.
2. The paper is well-organized and the key idea is clearly delivered. Open-source code is also provided for reproducibility.

**Weaknesses:**

1. Using an incorrect format (ICLR 2024) for submission.
2. Typos (Also, please add indexes to each equation to make it easy to be referred to):
- In the first equation about the definition of $L(\theta)$, chapter 3.1, there is an extra ")".
- In the last equation of chapter 3.1, $n$ is missing on the top of the $\Sigma$ colored in red.
3. There's not much originality in the key idea of simply combining the model-generated data with the real data to augment the training process, which has already been used in previous works [1] and other model-based reinforcement learning algorithms [2-3]. As the distribution shift problem has been widely researched in offline RL, it's not surprising that a similar problem will occur in off-policy RL. Although the authors provide theoretical analysis to show the instability brought by target policy action selection, more profound results are expected, such as how will the model error affect the performance and when to trust model-generated data in off-policy RL.
4. The algorithm is mainly based on TD-MPC2, while several components are changed. While the proposed MAD-TD is compared with the original TD-MPC2, the influence of these changes on performance has not been clearly explained or demonstrated.

[1] Lu, Cong, Philip Ball, Yee Whye Teh, and Jack Parker-Holder. "Synthetic experience replay." Advances in Neural Information Processing Systems 36 (2024).
[2] Sun, Yihao, Jiaji Zhang, Chengxing Jia, Haoxin Lin, Junyin Ye, and Yang Yu. "Model-Bellman inconsistency for model-based offline reinforcement learning." In International Conference on Machine Learning, pp. 33177-33194. PMLR, 2023.
[3] Rigter, Marc, Bruno Lacerda, and Nick Hawes. "Rambo-rl: Robust adversarial model-based offline reinforcement learning." Advances in neural information processing systems 35 (2022): 16082-16097.

**Questions:**

How will the model data proportion affect the result? I suggest the author provide some preliminary results or insights regarding the impact of model data proportion and conduct ablation studies, if necessary, to show the trade-off on this key parameter in your method. Besides, is it possible to develop a self-adaptive mechanism to control the model data proportion during the training to enhance the final performance? To my understanding, the model error could be large at the beginning of the training process. Therefore, using fewer model-generated data at first and gradually adding the proportion as the world model converges might be helpful.

---

### Official Review · Reviewer_jfC9 · 2024-10-31

**Soundness:** 3
**Presentation:** 3
**Contribution:** 3
**Rating:** 8
**Confidence:** 4

**Summary:**

The authors address the instability problem in off-policy reinforcement learning with the Replay Ratio. Building on previous results showing that neural networks trained on a specific set of experiences with the UTD lead to various instabilities in training, they show that the current solutions mitigate some of the problems to a greater or lesser extent, e.g., feature normalization mitigates the overestimation problem. Still, it does not allow the model to generalize to unseen actions. Therefore, they propose using a world model to augment a small part of the experiences from the replay buffer with its help.

**Strengths:**

1. Authors propose a method that works effectively with high UTD without resets.
2. The topic is relevant.
3. The paper is well-written.

**Weaknesses:**

1. The results look pretty good, but then there is a question about robustness. Authors sometimes refer to [1], but the same paper points out that different effects can be obtained in RL in different environments. I agree with the theses in the manuscript, but I think it is valuable to validate these strong results on other benchmarks so that scientists in the future will know to what extent this is a general solution. Please see question 1 for more details.
2. Minor: Equations are without numeration.
3. Lack of an analysis of the impact of the percentage of augmented samples on performance. Could you provide plots showing performance across a range of alpha values (e.g., 1%, 5%, 10%, 25%, 50%) on a subset of representative tasks? This would give a clearer picture of the method's sensitivity to this hyperparameter.


[1] Nauman, M., Bortkiewicz, M., Miłoś, P., Trzcinski, T., Ostaszewski, M., & Cygan, M. Overestimation, Overfitting, and Plasticity in Actor-Critic: the Bitter Lesson of Reinforcement Learning. In Forty-first International Conference on Machine Learning

**Questions:**

1. Referring to Weakness 1.: What would the results be on the other environments or benchmarks, like Meta-World (especially environments like Stick-pull, coffee-push/pull, assembly, and others) or MyoSuit?
2. Is there any advantage to using AR=2 rather than AR=1? Could you extend Figure 5 with AR=1 and 2 million time steps?

---

### Official Review · Reviewer_6ahP · 2024-11-01

**Soundness:** 3
**Presentation:** 3
**Contribution:** 3
**Rating:** 8
**Confidence:** 2

**Summary:**

In this paper the authors consider the reasons for RL to become unstable in UTD situations and they suggest a fix for this which they call MAD-TD. Basically, the data is augmented by that generated from a model to enable stabilisation. The authors then illustrate the success of their results using experiments.

**Strengths:**

I found this an interesting paper and the authors approach seems to generate useful results. The paper is well written and the presentation allows the reader to understand the rationale behind most of the work.

The paper combines some mathematical and intuitive insight into the stability problem. It then uses this insight as motivation for their new approach MAD-TD which seems to show some success based on the experimental results.

I thought the authors gave quite a balanced presentation of the strengths, but also possible weaknesses of their work, which is commendable.

**Weaknesses:**

The main limitation of the paper is precisely that which the authors point out themeselves i.e. that the assumption that a sufficiently high fidelity of the model can be learned online is valid. This is necessary as the "augmented data" is generated from this model. However, the authors have been quite up-front with this and, despite this short-coming, their results seem to show success with their MAD-TD approach.

Another shortfall of the techniques is that of course, the main "proof" of the results is via experiments, although some mathematical insight is given. In other words, the authors do not prove that their MAD-TD approach is able to prevent instability, they observe, through their experiments that it seems to do well.

**Questions:**

1. In the un-numbered equation on Page 3 (Please number equations!) a certain matrix is partitioned into a positive definite and a non-positive definite term. I was not quite sure what to make of the discussion below this. It of course, follows that if \gamma is shown sufficiently small, or that the first term strongly dominates the second. I would appreciate more discussion of this.

2. It wasn't clear to me what sort of guarantees we expect with the authors approach - it seemed to be more of a "it seemed to work most of the time"....except when it didn't. I didn't quite understand the insight about why in certain circumstances the approach didn't work. Is that related to the assumption that a sufficiently high fidelity model cannot always be learned? Some more insight would be useful.

---

### Official Review · Reviewer_zdTG · 2024-11-02

**Soundness:** 2
**Presentation:** 3
**Contribution:** 2
**Rating:** 8
**Confidence:** 3

**Summary:**

This paper addresses the challenge of unstable training in off-policy reinforcement learning (RL) methods when the update-to-data ratio is high. The authors identify the root cause of this instability as the difficulty in learning accurate value functions from limited data. To mitigate this issue, they propose a novel approach called Model-Augmented Data for Temporal Difference learning (MAD-TD). MAD-TD leverages model-generated data to improve the accuracy of value functions on unobserved on-policy actions, thereby stabilizing training even at high update ratios. They empirically show that MAD-TD achieves competitive performance on tasks from the DeepMind control suite.

**Strengths:**

1. The paper tackles an interesting and critical problem setting in sample-efficient reinforcement learning (RL), which has significant implications for many real-world applications where data collection is costly or time-consuming.
2. The authors provide both empirical evidence and theoretical analysis to demonstrate the importance of addressing incorrect Q-value learning in off-policy RL with limited samples, making a compelling case for their proposed solution.
3. The proposed method, MAD-TD, demonstrates competitive performance compared to existing baselines on challenging DeepMind Control (DMC) tasks, showcasing its potential as a viable solution for improving sample efficiency in RL.

**Weaknesses:**

1. Limited Baseline Comparison: The paper only compares the proposed method, MAD-TD, against two baselines (BRO and TD-MPC) and their variants, which may not be sufficient to demonstrate its performance comprehensively. A more extensive comparison with other state-of-the-art methods would strengthen the paper's claims.
2. Lack of Ablation Study: Although the authors outline critical design choices in Section 4.1, they do not conduct an ablation study to investigate the importance of these choices. This omission makes it difficult to understand the individual contributions of each design element to the overall performance of MAD-TD.

**Questions:**

1. Have you explored different values for the percentage of model generated data in MAD-TD?
2. How does the performance of MAD-TD vary with different levels of world model accuracy? Have you conducted any ablation studies on less accurate models to understand how sensitive MAD-TD is to the world model quality?

---

### Meta-Review · Area_Chair_CAGy · 2024-12-21

**Metareview:**

This paper addresses the issue of overestimation in value estimates that preclude high update-to-data ratio. The paper proposes to use a world model and update the value using on-policy data generated by the world model to correct this overestimation. The effectiveness of their approach is shown on hard continuous control Dog tasks from the DM control suite.

The approach is intuitive, and the results are quite promising. The paper contributes significantly to understanding and addressing the challenges of high update-to-data ratios in RL. Likewise, I recommend accepting the paper.

**Additional Comments On Reviewer Discussion:**

The reviewers have clear consensus in favor of the paper. Any concerns raised were sufficiently resolved through extensive discussion and addition of results during rebuttal. However, it should be noted that the work relies on the assumption that a sufficiently accurate world model can be learned online, which may not be true in many cases.

---

### Decision · Program_Chairs · 2025-01-22

Accept (Spotlight)